# Design and Manufacturing of Antibacterial Electrospun Polysulfone Membranes Functionalized by Ag Nanocoating via Magnetron Sputtering

**DOI:** 10.3390/nano12223962

**Published:** 2022-11-10

**Authors:** Noemi Fiaschini, Chiara Giuliani, Roberta Vitali, Loredana Tammaro, Daniele Valerini, Antonio Rinaldi

**Affiliations:** 1NANOFABER S.r.l., Via Anguillarese 301, 00123 Rome, Italy; 2SSPT-PROMAS-MATPRO, ENEA—Italian National Agency for New Technologies, Energy and Sustainable Economic Development, Via Anguillarese 301, 00123 Rome, Italy; 3SSPT-TECS-TEB, ENEA—Italian National Agency for New Technologies, Energy and Sustainable Economic Development, Via Anguillarese 301, 00123 Rome, Italy; 4SSPT-PROMAS-NANO, ENEA—Italian National Agency for New Technologies, Energy and Sustainable Economic Development, Piazzale E. Fermi, 1, Portici, 80055 Napoli, Italy; 5SSPT-PROMAS-MATAS, ENEA—Italian National Agency for New Technologies, Energy and Sustainable Economic Development, S.S. 7 Appia, km 706, 72100 Brindisi, Italy

**Keywords:** antimicrobial, polysulfone (PSU), silver nanoparticles, nanocoating, electrospinning, magnetron sputtering, antibacterial, biomedical, water filtration, antifouling

## Abstract

Antibacterial properties of engineered materials are important in the transition to a circular economy and societal security, as they are central to many key industrial areas, such as health, food, and water treatment/reclaiming. Nanocoating and electrospinning are two versatile, simple, and low-cost technologies that can be combined into new advanced manufacturing approaches to achieve controlled production of innovative micro- and nano-structured non-woven membranes with antifouling and antibacterial properties. The present study investigates a rational approach to design and manufacture electrospun membranes of polysulfone (PSU) with mechanical properties optimized via combinatorial testing from factorial design of experiments (DOE) and endowed with antimicrobial silver (Ag) nanocoating. Despite the very low amount of Ag deposited as a conformal percolating nanocoating web on the polymer fibers, the antimicrobial resistance assessed against the Gram-negative bacteria *E. coli* proved to be extremely effective, almost completely inhibiting the microbial proliferation with respect to the reference uncoated PSU membrane. The results are relevant, for example, to improve antifouling behavior in ultrafiltration and reverse osmosis in water treatment.

## 1. Introduction

Antibacterial and antiviral properties of materials and components are fundamental to many industrial fields, such as health and water treatment, and are pervasive to the transition towards a circular economy and societal security. In general, an antimicrobial agent is defined as a natural or synthetic substance that kills or inhibits the growth of pathogenic microorganisms such as bacteria, fungi, and algae.

Some of the commonly used antimicrobial agents, for example, include metal ions, polymers, antimicrobial peptides, quaternary ammonium compounds, naturally derived antimicrobials, and so on.

In particular, antibacterial properties of nanoparticles (NPs) are widely proven and shown to provide a significant response even at low concentrations. Metal-based nanoparticles (NPs) have been extensively investigated for a set of biomedical applications and employed as antimicrobial agents [1]. The metals used are almost exclusively heavy metals, such as Ag and Cu [2]. Ag has been utilized as an antimicrobial agent for several millennia. Since Hippocrates prescribed the use of Ag to treat ulcers [3] and as nanotechnology became an established engineering discipline [4], Ag NPs have become a staple tool in antimicrobial applications, especially for combatting antibiotic-resistant bacteria and nosocomial infections [5].

Engineered materials can acquire specific antimicrobial properties by means of surface treatments and nanotechnologies, particularly nanocoating processes [6] in association with electrospinning [7,8].

Nanofibers produced by electrospinning have been investigated intensively over the past decade to obtain innovative membranes with excellent properties, such as high surface area to volume ratio, high porosity, and flexibility [9]. At the same time, nano-based antimicrobial coatings are increasingly used to impart a range of functionalities, either a broad-spectrum protection against various microbes or a narrow-spectrum protection against a specific sector of microbes [10]. This possibility was highlighted to society at large by the recent COVID-19 global pandemic [11], with one of the many associated formidable challenges being the design of new, more effective antiviral barriers for protective devices, face masks, and garments to be sourced in quantities of billions [12].

The deployment of Ag-based systems is certainly one of the most recurring options in current products and a favourite topic of debate in the research community [13,14,15,16,17].

In a prior work [18], our group investigated the possibility of applying a nanocoating of Ag via sputtering onto electrospun membranes endowed with antibacterial properties. The findings indicated that it was possible to fine-tune the coating process to be applicable for use on delicate membranes of polycaprolactone (PCL), a soft-polymer with low melting point (55–60 °C). The resulting ultra-thin Ag nanocoating was capable of delivering a very effective antibacterial barrier vs. *Escherichia coli* despite the minimum amount of this precious metal, making it potentially cost-effective and industrially attractive.

In the present study, we expand our previous work by focusing on electrospun membranes of polysulfone (PSU) and providing a methodological approach by design of experiment (DOE) to optimize the PSU substrate by maximizing its mechanical properties before submitting it to the Ag coating processes. PSU is an amorphous thermoplastic polymer known for its relatively high strength, high temperature stability (melting point ca. 180 °C), low creep, good electrical characteristics, and resistance to many solvents and chemicals. PSU finds applications in several fields, such as medical, electrical, electronic, aerospace, automotive, coating, etc. Furthermore, it is one of the industry standard materials used in ultra-filtration and reverse osmosis membranes in water treatment. For such an application, antibacterial functionalization is a crucial area of research and innovation to foster antifouling and increase the lifespan of the PSU separation membrane [19,20,21].

Recently, PSU membranes have been combined with Ag nanoparticles to add antibacterial behavior and control the membrane wettability properties for usages in water filtration and other possible applications [22,23,24,25,26]. However, only in a few of them [25,26] the PSU membranes are fabricated by the electrospinning method, which can present numerous advantages, as described above. Additionally, in all those reports, the Ag functionalization was achieved by blending the Ag nanoparticles with the PSU polymer into the initial electrospinning solution. On the contrary, the post-processing of the electrospun bare PSU fibers by subsequent Ag coating can provide several benefits, as extensively described in our prior work [18]; briefly, overcoming possible wettability mismatch between nanoparticles and polymer in the electrospinning solution; preservation of the hydrophobic character in the composite material; reduced use of harmful precursors; presence of the active antibacterial agent mainly on the surface, thus maximizing its direct exposure to the surrounding area even at low load; adherence/resistance of the coating; selective surface functionalization; industrial scalability; and adaptability to a wide range of materials (both in terms of materials that can be functionalized and of antimicrobial agents that can be deposited).

The present paper reports the manufacturing and characterization of electrospun PSU meshes with surfaces functionalized by Ag-nanocoating using the magnetron sputtering technique, thus retaining the advantages of both techniques, as described above. Structural and physical properties (mechanical and wettability) were investigated and correlated to the nanofiber structure. Moreover, the analysis of the results showed that the addition of the Ag nanocoating created a surface with significant antibacterial activity, making these functionalized PSU non-woven mats attractive as antimicrobial membranes for water treatment applications.

## 2. Materials and Methods

### 2.1. Electrospinning of PSU Scaffolds and DOE Factorial Design Approach

PSU microfibrous sheets were produced via electrospinning using needle-technology electrospinning equipment (Fluidnatek LE100, Bioinicia, Spain) outfitted with a flat collector and two-axis emitter motion.

The electrospinning process is highly dependent on process parameters (*X_s_*), which determine the properties (*Y_s_*) of the resulting membrane [27]. The solution was prepared and processed on the basis of the process recipe of a commercially available product (PSU-NBARE ™ series, NANOFABER S.r.l., Italy), kindly provided by the Company NANOFABER S.r.l. (Rome, Italy). A 24% *w/w* solution of PSU polymer dissolved in *N*-Methyl-2-pyrrolidone (NMP) was prepared by pre-drying pure PSU (granule 2 mm, GradeUdel^®^ P 1700, Goodfellow, Huntingdon, UK) at 50 °C for five hours and then stirring them until complete dissolution in NMP (100% purity, VWR, Radnor, PA, USA) at room temperature. The initial original recipe was modified for optimization purposes, in this study, following a combinatorial full-factorial 2^3^ approach from DOE [28], which was implemented to produce a small design with eight runs to investigate the effect of the three process parameters reported in Table 1 (everything else being constant) on the mechanical properties of the electrospun mat listed in Table 2. Ambient conditions were finely controlled, such that the relative humidity and the temperature in the process chamber were kept, respectively, at average values of 47.0% (with a mean square error (MSE) of ±1.4%) and of 25.1 °C (with a MSE of ±0.2 °C) throughout the eight runs.

The optimization goal was to select the best PSU electrospun membranes in terms of higher mechanical strength (UTS) on which to apply the antibacterial coating. The first- and second-best samples were submitted to coating processing. Stiffness properties were examined, as well, for completeness. The DOE approach provides a methodological tool to map the mechanical performance vs. the process domain via linear models.

We recall that for any given Y output, the following linear model is fitted to the experimental data to obtain a linear “ordinary least square” estimate for (𝑋_1_, 𝑋_2_, 𝑋_3_) in coded variable, according to the equation:(1)$$y=C_{0}^{*}+C_{i}^{*}x_{i}^{*}+C_{12}^{*}x_{1}^{*}x_{2}^{*}\;\;\;\left(i=1\ldots 3\right)$$
obtained through linear transformations from natural variables *X_s_*:(2)$$x_{i}^{*}\left(x_{i}\right)=\frac{xi-\bar{x}i\;}{\left(xHIGH-xLOW\right)/2}\;\;\;\left(i=1\ldots 3\right)$$
with HIGH and LOW levels mapping to +1 and −1 (the “*” superscript is dropped, hereafter, for readability). Equation (1) includes primary coefficient terms relating “primary effects” {𝐶_1_, 𝐶_2_, 𝐶_3_} and other terms linked to the “interaction” between pairs of them, up to order two, {𝐶_12_, 𝐶_13_, 𝐶_23_}. Using coded variables is convenient for assessing, at a glance, the relative importance of one regressor (i.e., a main parameter or an interaction effect) over the other ones.

The mechanical tests for the determination of *Y*_1_ and *Y*_2_ were carried out on thin strips, as reported in the next section. The analysis of variance (ANOVA) was carried out via statistical software MINITAB© (Minitab LLC, State College, PA, USA). All subsequent processing (sputtering), testing (cell culture), and characterization (contact angle, SEM, XRD, FTIR) tasks were performed on PSU die-cut disks with a diameter of 15 mm, chosen to fit tightly at the bottom of a well in a standard 24-multiwell culture plate.

### 2.2. Mechanical Tests

The tensile properties were evaluated by a macro-tensile loading frame (MICROTEST 200 N, DEBEN, Suffolk, UK) equipped with a 200 N load cell. Rectangular strips were cut and mounted in the loading frame (Figure 1), with a free gauge length of *l_0_* = 15 mm and a uniform width (*w*) between 5 and 7 mm. The sample thickness (*s*) varied between 24 and 113 μm, as measured by SEM imaging of the cross-section. Load vs. cross-head displacement data was recorded during uniform tensile tests conducted at a strain rate of 0.1% (i.e., movable cross-head moving at 1 mm/min) at room temperature. The engineering stress (σ, MPa) vs. strain (ε, %) curves in Figure 1 were calculated by dividing, respectively, the applied load by the apparent cross-sectional area (A_0_ = *w*
×
*s*) and the cross-head displacement by *l_0_*, corrected for elongation amount, needed to fully stretch the strip. The UTS (*Y*_1_) was measured as maximum tensile stress before rupture or onset of unloading with marked necking, whereas elastic modulus (*Y*_2_) was estimated from the slope of the linear fit in elastic region for each stress-strain curve.

### 2.3. Sputter-Deposition of Ag Nanocoatings

PSU disks were coated by DC sputtering (equipment Kenosistec, Italy, mod. KS 1000 SEA) from a circular Ag target (Testbourne Ltd., Basingstoke Hampshire, UK) with diameter of 3 inches (76.2 mm) and purity of 99.99%, placed in a vacuum chamber in bottom-up configuration with confocal geometry (target inclination of 30° with respect to the substrate holder and target-substrate distance of 140 mm). Before the deposition, the vacuum chamber was evacuated to a base pressure of about 6 × 10^−6^ Pa. Then, Ar gas (99.9999% purity) was flowed at 20 sccm into the chamber and the working pressure was set at 2 Pa. The DC power was set at 30 W, the target surface was cleaned through pre-sputtering for 5 min, and then Ag deposition was carried out for 3′30″ onto the substrates placed on the holder rotating at 10 rpm, providing a deposition rate of about 1.8 nm/min. The deposited Ag coating then resulted in a nominal thickness of about 6 nm, corresponding to a silver load of about 4.5 µg/cm^2^, as estimated in our previous analogous work [18], and corresponding to a weight ratio between Ag and PSU of about 0.1–0.2%wt.

### 2.4. Scanning Electron Microscopy (SEM) and Energy Dispersive X-ray Spectroscopy (EDS)

The morphological properties of PSU electrospun material were examined in pristine and Ag-coated conditions by using a field emission gun scanning electron microscope Leo 1530 model (ZEISS, Jena, Germany) working at low voltage (i.e., 2 kV) to avoid charging effects and damage to the dielectric polymer from overheating. As mentioned in Section 2.2, membrane thickness was also evaluated via SEM on the sample cross section. Energy- dispersive X-ray spectroscopy (EDS) measurements were taken using an X-MAX detector (AZTEC, Oxford, UK). PSU sheets were examined to ascertain and map the distribution of Ag on coated microfibers of the PSU samples. For EDS, the operating voltage of the SEM was raised to 5 kV to observe Ag peaks in the EDS spectrum.

### 2.5. Attenuated Total Reflectance-Fourier Transform Infrared Spectroscopy (ATR-FTIR)

The electrospun PSU membranes were characterized before and after Ag-coating deposition by *ATR-FTIR* spectroscopy. The spectra were collected using a Nicolet iS50 spectrometer (Thermo Fisher Scientific, Waltham, MA, USA) equipped with an ATR accessory. The measurements were recorded using a diamond crystal cell ATR, typically using 32 scans at a resolution of 4 cm^−1^. The samples were all measured under the same mechanical force pushing the samples in contact with the diamond crystal. No ATR correction has been applied to the data.

### 2.6. X-ray Diffraction (XRD)

To study the crystalline structure of the uncoated and Ag-coated PSU membranes, X-ray diffraction (XRD) patterns were collected with a Philips-X′Pert MPD X-ray diffractometer (Philips, now Panalytical, Malvern, UK), operating at 40 kV and 40 mA, in the range of 2θ = 10–80°, at a scanning rate of 0.005°/s with a step size of 0.05°, equipped with a Cu-sealed tube using Kα radiation (λ = 1.54056 Å). The data were analysed using X′Pert Quantify software.

### 2.7. Contact Angle Measurements (WCA)

The water contact angles of the PSU electrospun sheets were measured by means of an optical contact angle measuring system, OCA 20 (DataPhysics, Filderstadt, Germany), equipped with SCA 202 software, using ultra-pure water, and operating at room temperature. Determinations were made using the sessile drop method with 1 μL droplet volume and a deposition rate of 1 µL/s. Ten measurements were carried out for each sample from different locations and the average value reported with its standard deviation (SD). The same number of images were captured.

### 2.8. Antibacterial Tests

The die-cut PSU disks of 15 mm diameter fitted in the 24-multiwell plates were used for the biological tests. *Escherichia coli* (*E. coli*) stock cultures kept at −80 °C in 10% (wt./vol.) glycerol (Merk Life Science S.r.l., Milan, Italy) were inoculated into 5 mL of Luria–Bertani (LB) broth (Merk Life Science S.r.l, Milan, Italy) and incubated at 37 °C O/N before their use in experiments. *E. coli* was pre-inoculated aerobically for 16 h at 37 °C in LB medium, with constant shaking at 250 rpm. The day of the test, the bacteria were diluted and grown until OD600 was 0.027, about 2.0 × 10^3^ colony forming units/mL (CFU/mL). Then, the bacteria were placed in a multiwell plate with 24-wells in the presence of uncoated PSU samples (blank disks) or Ag-coated samples, and incubated at 37 °C under constant agitation at 50 rpm. A control with *E. coli* alone was also inserted. At times *t* = 1, 2, and 3 h, 10 μL of each sample were diluted in phosphate-buffered saline (PBS) (Euroclone, Milan, Italy) (serial dilution from 10^−1^ to 10^−4^) and 10 μL of each dilution were distributed on LB agar dishes (15 g L^−1^ agar) (Panreac Química SLU, Barcelona, Spain) and incubated for 18 h at 37 °C. Each plating was performed in triplicates. Subsequently, the number of CFU/mL was quantified for each condition. The experiment was repeated twice.

### 2.9. Statistical Analysis of Antibacterial Tests

To determine the effect of the Ag nanocoating on the antibacterial properties, two approaches were adopted, as detailed below.

**Method** **1.**The CFU for each treatment and for each time-point (i.e., 1 h, 2 h, 3 h) was normalized by the seeding density at time zero, i.e., 2.0 × 10^3^ CFU/mL. The two-sample *t*-test was used to carry out pairwise comparisons between mean values of different treatments or time-points in the antibacterial tests panel. The difference between any given pair of mean values $\langle y_{1}\rangle \;\mathrm{and}\;\langle y_{2}\rangle \;$with mean square errors $S_{1}$ and $S_{2\;}$ was significant if
(3)$$\left|t_{0}\right|>t_{\alpha /2,2}\;$$
where $t_{\alpha /2,2}$ is the threshold value for *n* observations depending on significance α. Since triplicates were conducted and they were repeated twice, by computing <*y*_0_> of the treatment as the mean value of each triplicate, it is *n* = 4, and the value $t_{\alpha /2,2}=2.132\;$was taken for α = 0.05.Likewise, *t*_0_ was computed from the general formula:
(4)$$t_{0}=\frac{{\bar{y}}_{1}-{\bar{y}}_{2}}{\sqrt{\frac{S_{1}^{2}}{n_{1}}+\frac{S_{2}^{2}}{n_{2}}}}$$
where the *S*_1_ and *S*_2_ are obtained as the mean square error for each pair of repeated triplicates. The mean difference analysis based on Equations (3) and (4) was implemented on a spreadsheet and augmented with plots from statistical software JMP-pro (SAS Institute, Cary, NC, USA).

**Method** **2.**For a more immediate view of the influence of the different samples (uncoated and Ag-coated PSU disks) on the microbial growth over time, the bacterial concentration (CFU/mL) measured after contact with the disks at the different time intervals was firstly normalized with respect to the concentration of the *E. coli* control culture at the same intervals. This normalized bacterial population (*N_t_*) is thus obtained by:
(5)$$N_{t}=\frac{T_{t}/T_{0}}{C_{t}/C_{0}}$$
where *T_t_* and *C_t_* are the microbial concentrations detected at the different time intervals (*t*) in contact with the test disks (*T*) and for the control culture (*C*), respectively, while *T*_0_ and *C*_0_ are the corresponding initial concentrations at *t* = 0. All data were calculated as the mean value ± SD, as resulting from the different measurement replicas.In order to quantify the effective bacterial reduction induced by the addition of the Ag nanocoating on the PSU material, the population measured on the coated PSU-Ag disks was divided by the corresponding population on the uncoated PSU substrates. In this way, the residual bacterial population (*R_t_*) on the coated samples can be expressed as a percentage, with respect to the corresponding uncoated substrates at each considered time (*t*):
(6)$$R_{t}\;\left(\%\right)=\frac{N_{t\_coated}}{N_{t\_uncoated}}\times 100$$The data were then fitted through an exponential decay model with formula $y=e^{-\alpha t}$, where *y* is the residual bacterial population, *t* is time, and α is the decay rate, using the Levenberg–Marquardt algorithm for the least squares regression.

## 3. Results

### 3.1. Electrospinning, Mechanical Characterization, and DOE Results

Eight batches of membranes were electrospun and characterized. Process parameters and results are reported in Table 3. The two best-performing PSU-based electrospun materials in the study were identified as DOE3 and DOE7, respectively, ranked first and second by UTS.

Regression models were computed for both UTS and E using MINITAB©. The fitting was satisfactory, with coefficients of determination R^2^ > 99%, indicating a good description of the data variability (Table 4). Limited to *Y*_1_, the corresponding regression model equation is the following:*Y*_1_: UTS (MPa) = 8.362 + 3.488 FR(mL) − 1.138 Vi(kV) − 2.263 FR(mL) × Vi(kV) − 1.463 FR(mL) × d(cm) + 1.913 Vi(kV) × d(cm)(7)
and the main results of ANOVA are provided in Table 5. Figure 2 and Figure 3 display, respectively, the relative significance of main effects and interaction through the Pareto chart and normality of residuals.

### 3.2. Characterization of the Samples before and after the Ag Nanocoating Process

Some representative samples of electrospun DOE3 and DOE7 PSU disks before and after sputter-deposition of Ag are shown in Figure 4.

The morphology of the PSU disks before and after the Ag treatment was investigated using SEM. Micrographs at low magnification (Figure 5) show that the membranes are composed of a dense net of randomly oriented fibers. The diameter of the fibers is broadly distributed from 100 nm or less to a few microns. The diameter distributions, as determined by fiber counting on SEM micrographs, were 0.82 ± 0.45 µm and 2.24 ± 1.27 µm for DOE3 and DOE7, respectively (mean values ± MSE).

The micrographs at high magnification in Figure 6 clearly reveal the presence of the Ag nanocoating on the surface of the PSU fibers, confirming the effectiveness of the coating process. Such a coating appears as a conformal nanoscale percolating network. The analysis demonstrates that the deposition process allowed a uniform coverage of the polymer fibers, also resulting in a high conductivity of the coated samples during SEM inspections, and moreover, that this process did not induce any significant modification of the fiber bundle. Thus, the proposed coating process is non-destructive and suitable for such materials, confirming our previous results on other kinds of soft polymer electrospun membranes [18]. EDS analysis at high resolution confirmed the chemical signature of Ag on the coated fibers (Figure 7).

As addition to morphological evidence, FTIR analysis also indicated that the Ag deposition process did not induce important alterations and degradation of the polymer molecular structure. FTIR spectra of the electrospun PSU samples before and after the sputter-deposition reported in Figure 8 show that there was no change of the polymer absorption bands after the coating process.

The X-ray diffraction spectra for uncoated PSU and Ag-coated PSU electrospun membranes are shown in Figure 9. The XRD pattern of polysulfone exhibited a broad peak centered at around 2θ = 17.9°, which indicated an almost amorphous structure. The diffraction peak at 38.1°, which corresponds to the (111) crystalline plane of the silver particles [29,30], was present throughout all coated samples together with the broad pattern associated with the PSU amorphous phase.

The results reported above indicate that the proposed coating processing, as optimized in our studies, can be considered substantially mild and potentially applicable on a wide range of soft materials in engineering.

### 3.3. Contact Angle Measurements

From a functional standpoint, the application of the Ag nanocoating was proven to modulate the hydrophobicity of the substrate membrane based on the effects on contact angles measured in wettability tests.

From the data shown in Figure 10, it clearly emerges that the Ag coating influences the wetting behaviour of water droplets on the PSU electrospun substrates, leading to an increased hydrophobicity, in accordance with our previous findings [18,29]. This effect is more pronounced in sample DOE3, where the contact angle is found to increase from about 112° for the uncoated PSU to about 142° for the Ag-coated PSU surface. The lower contact angle observed in sample DOE3, as compared to DOE7, could be attributed to the lower roughness of the sample surface [31], since the spread of a liquid droplet on a smooth surface is facilitated with respect to a rough surface where air can be trapped below the droplet and hinder its spread.

### 3.4. Antibacterial Tests

The antibacterial activity of the uncoated and Ag-coated PSU samples was determined against the Gram-negative bacteria *E. coli*. The samples were incubated with the microbial population, and the survival rate was determined at 1, 2, and 3 h using the count plate method and calculation of CFUs/mL to determine the antibacterial action of the samples. Figure 11 shows photographs of the Petri dishes obtained from control culture (*E. coli* in the absence of any mat) and from cultures with both PSU blanks (DOE3, DOE7) and PSU-Ag mats at different serial dilutions and different time points. Table 6 presents data obtained from the CFU calculations.

As expected, *E. coli* culture increased from the initial concentration of 2.0 × 10^3^ CFU/mL at *t* = 0 (see Section 2.8) to 7.3 × 10^6^ CFU/mL at 3 h. The bacterial population also increased in the uncoated PSU samples, reaching 6.6 × 10^6^ and 1.7 × 10^7^ CFU/mL in presence of DOE3 and DOE7, respectively. On the contrary, the addition of the Ag nanocoating on both PSU substrate types led to a strong reduction of bacterial growth, in accordance with our previous findings [18,29]. Significance was investigated and quantified in two ways. According to the matrix of Student t-values (Figure 12) constructed following method 1 in Section 2.9, the significance of Ag-coated vs. uncoated substrates was confirmed for both DOE3 and DOE7 at 3 h. Figure 13 shows, at a glance, the difference between treatments, also in comparison to the grand mean of pooled data.

Following method 2 described in Section 2.9, the bacterial population normalized with respect to the *E. coli* control culture is reported in Figure 14 (left panel), where the inhibitory effect in the coated samples is evident in comparison with the microbial growth on the uncoated PSU samples. The antimicrobial effect of the Ag nanocoating is more clear when calculating the residual bacterial population on the Ag-coated samples as a percentage with respect to the corresponding uncoated substrates, as demonstrated in Figure 14 (right panel).

Specifically, the Ag coating on the fibers induced a reduction of the bacterial population of about 58% and 84% after only 1 h on DOE3 and DOE7, respectively; 77% and 89% after 2 h; and 94% and 98% after 3 h, namely, a bacterial decrease of almost 2 Logs compared to blank PSU. Such a result is even more interesting when considering the very low content of Ag in our samples, i.e., an Ag/PSU ratio of about 0.1–0.2%wt. in our work vs. typical ratios in the range 0.3–5% employed in other previous works [22,23,25]. After the exponential decay fit of data in Figure 14 (right panel), DOE7-Ag resulted in a slightly faster antibacterial effect, with a bacterial decrease rate that was about 1.5 times higher than that of DOE3-Ag. As a possible factor inducing this behavior, the slightly bigger size of the fibers in DOE7-Ag could favor a wider contact area between the coated sample fibers and the bacterial cells, with a consequent higher transfer of active bactericide Ag ions from DOE7-Ag with respect to DOE3-Ag.

## 4. Conclusions

In the present work, we proposed the development of antibacterial materials based on electrospun PSU membranes modified with sputtered Ag nanocoatings. Different batches of PSU scaffolds were electrospun and characterized. The best-performing materials in terms of mechanical properties, i.e., samples DOE3 and DOE7, were selected for subsequent Ag deposition. All the samples were characterized before and after the Ag treatment. SEM and EDS revealed the presence of a uniform Ag nanocoating on the surface of the PSU membranes, confirming the effectiveness of the coating process. It is worth noting that no evidence of significant modification of the fibers was observed, thus demonstrating the non-destructiveness and suitability of the proposed nanocoating process for the functionalization of soft polymer electrospun membranes. The wettability tests revealed an increase in the hydrophobicity of the PSU-Ag composites. The antibacterial activity of the PSU-based samples was assessed against the Gram-negative bacteria *E. coli*. The antibacterial tests showed that, despite the very low amount of Ag deposited as nanocoating on the polymer fibers, this functionalization is able to strongly reduce the microbial growth. The obtained results in terms of polymer and electrospun scaffold properties and antimicrobial activity are promising for the application of the developed Ag-coated PSU electrospun membranes in ultra-filtration and reverse osmosis in water treatment. In this field, consideration of the antibacterial properties is a crucial requirement to prevent/reduce fouling and increase the service life of the PSU-based membranes.

## Figures and Tables

**Figure 1 nanomaterials-12-03962-f001:**
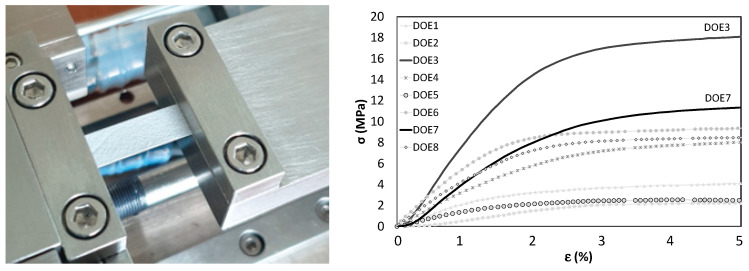
Mechanical characterization of electrospun self-standing membranes to determine formulation with best mechanical performance. Right: vice apparatus for microtensile tests (M200, Deben, UK) with 200 N load cell; left: stress vs. strain data for all DOE samples, with solid black lines highlighting DOE3 and DOE7 with highest UTS and chosen for the coating study.

**Figure 2 nanomaterials-12-03962-f002:**
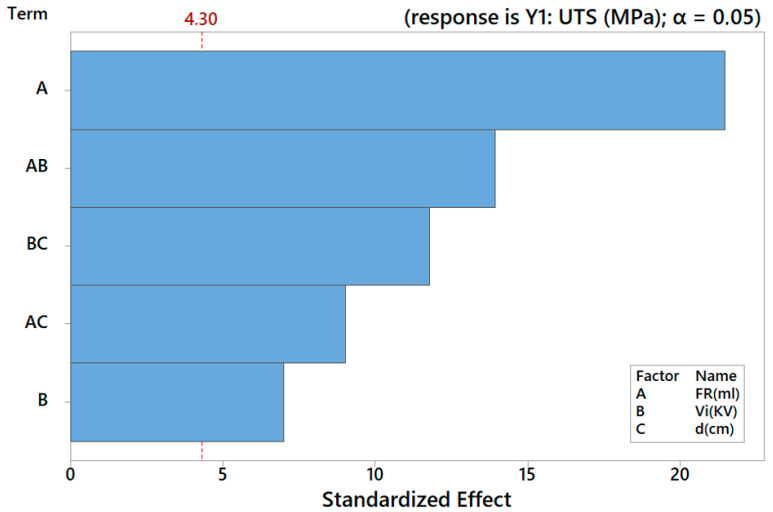
Pareto chart of the standardized effects for *Y*_1_.

**Figure 3 nanomaterials-12-03962-f003:**
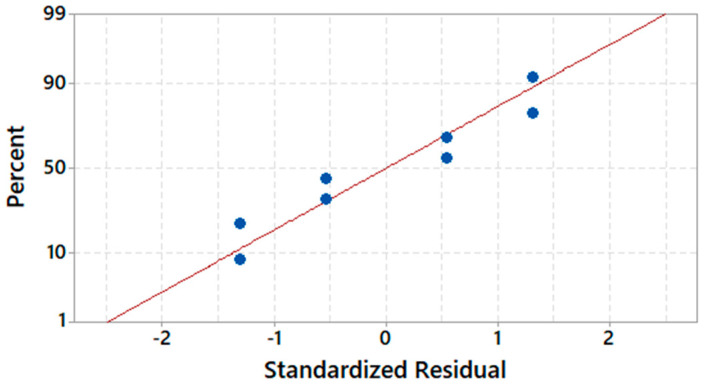
Normal probability plot of residuals for *Y*_1_.

**Figure 4 nanomaterials-12-03962-f004:**
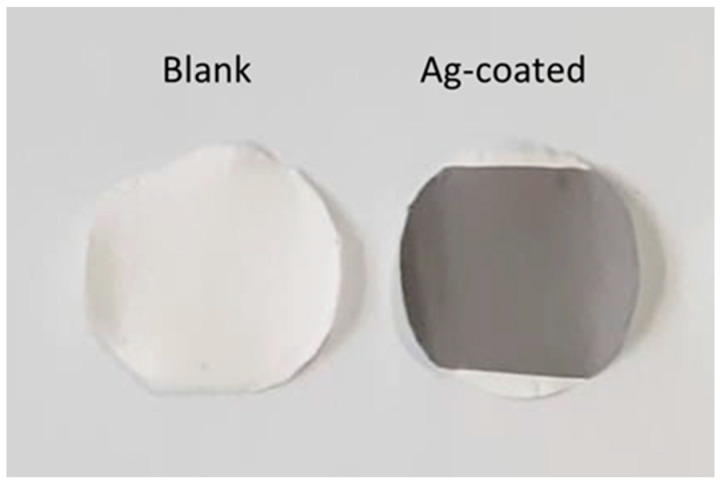
Representative samples of electrospun blank (**left**) and Ag-coated (**right**) PSU disks.

**Figure 5 nanomaterials-12-03962-f005:**
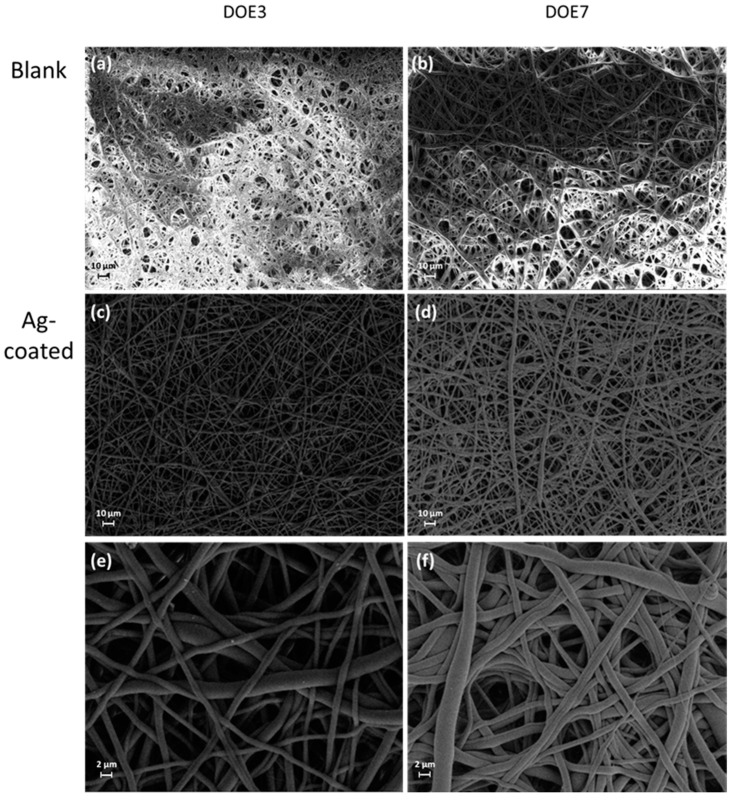
SEM micrographs at different magnification (from 500× to 5000×) of bare samples (**a**,**b**) and Ag-coated ones (**c**–**f**).

**Figure 6 nanomaterials-12-03962-f006:**
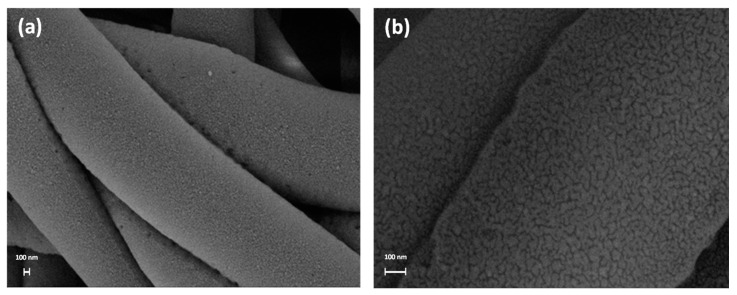
SEM micrographs at high magnification (**a**) 50K× and (**b**) 200K× of the Ag-coated PSU fibers.

**Figure 7 nanomaterials-12-03962-f007:**
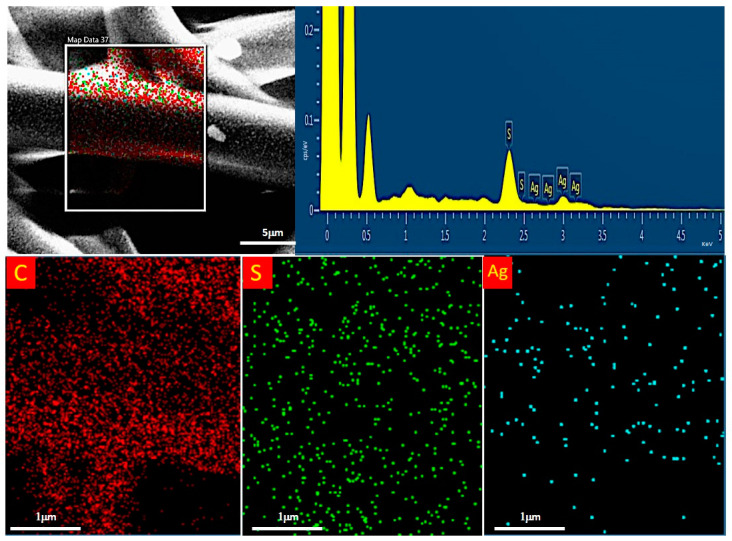
EDS analysis of the Ag-coated PSU fibers.

**Figure 8 nanomaterials-12-03962-f008:**
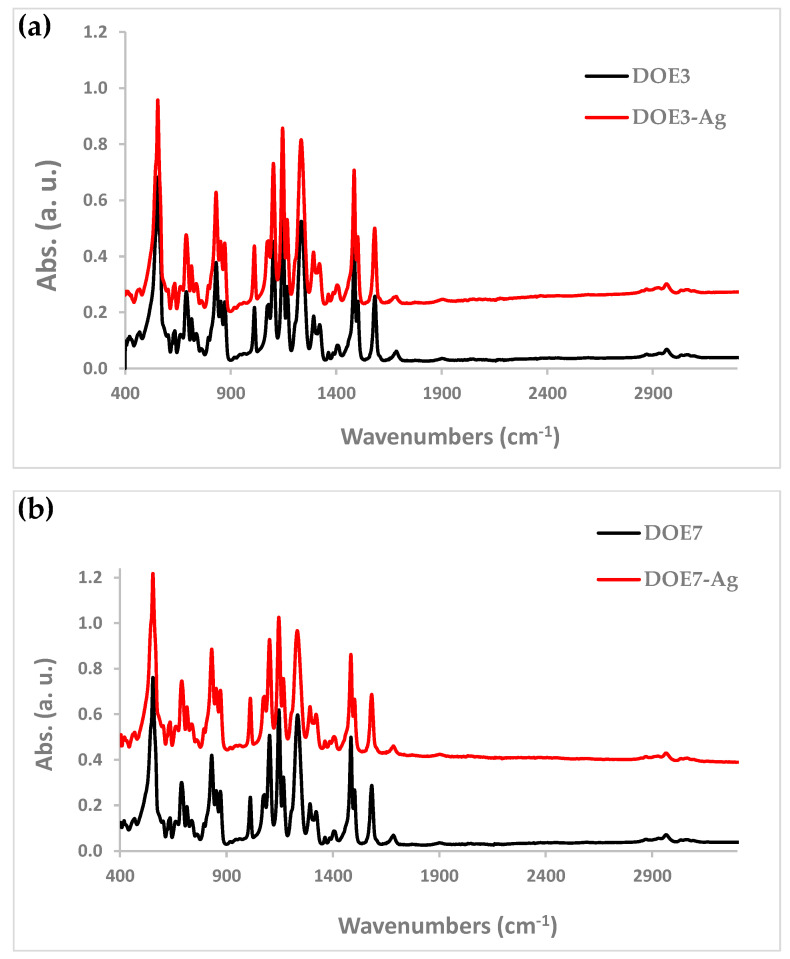
FTIR spectra of (**a**) DOE3 and (**b**) DOE7 samples before and after Ag deposition.

**Figure 9 nanomaterials-12-03962-f009:**
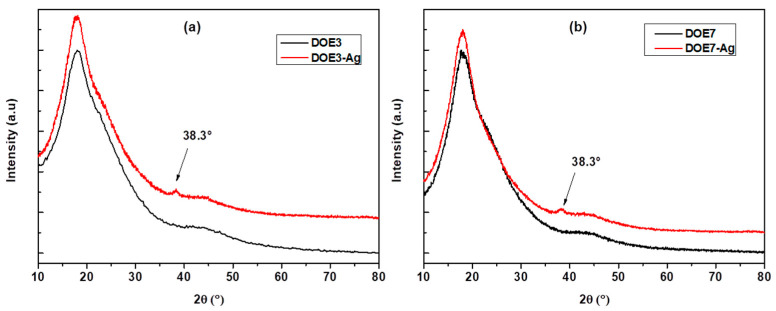
X-ray diffraction patterns of (**a**) DOE3 and DOE3-Ag, (**b**) DOE7 and DOE7-Ag.

**Figure 10 nanomaterials-12-03962-f010:**
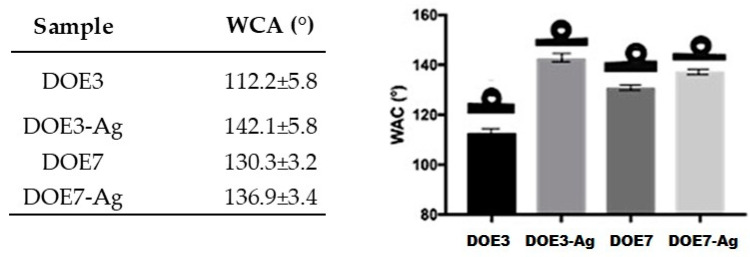
Water contact angle values (WCA) (mean ± standard deviation) on PSU and PSU-Ag substrates.

**Figure 11 nanomaterials-12-03962-f011:**
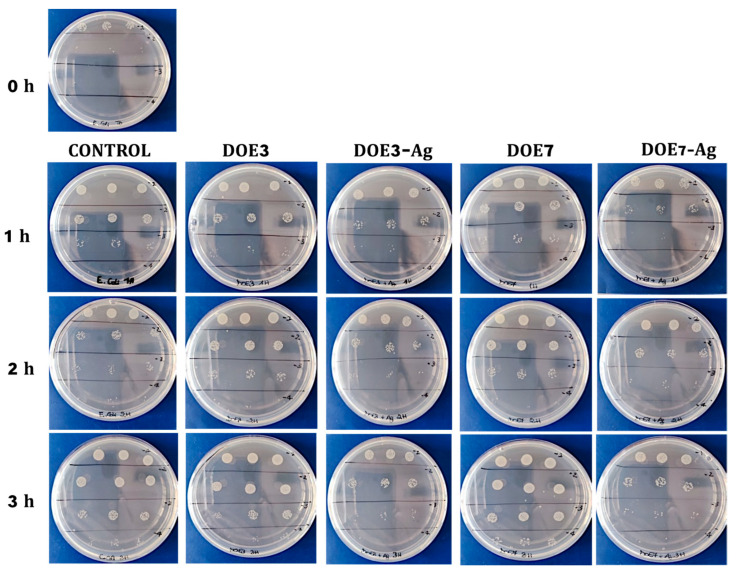
Photographs of Petri dishes obtained from *E. coli* culture without any mat (control) and cultures with blank PSU and PSU-Ag mats at different serial dilutions from 10^−1^ to 10^−4^ and different sampled times (0, 1, 2, and 3 h).

**Figure 12 nanomaterials-12-03962-f012:**
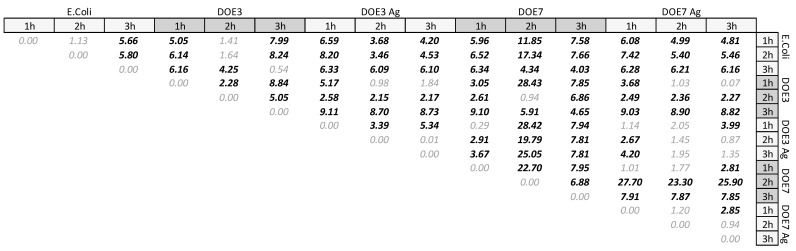
Matrix of Student t-values for mean-pair comparison.

**Figure 13 nanomaterials-12-03962-f013:**
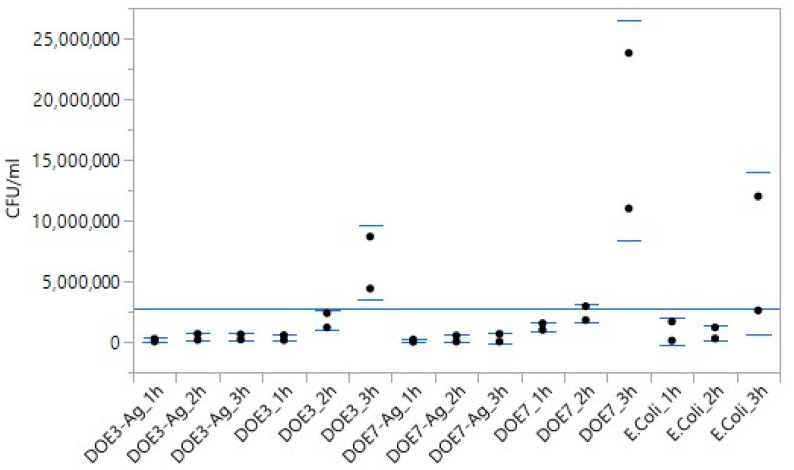
Plot of mean value of each treatment vs. grand mean of all pooled data.

**Figure 14 nanomaterials-12-03962-f014:**
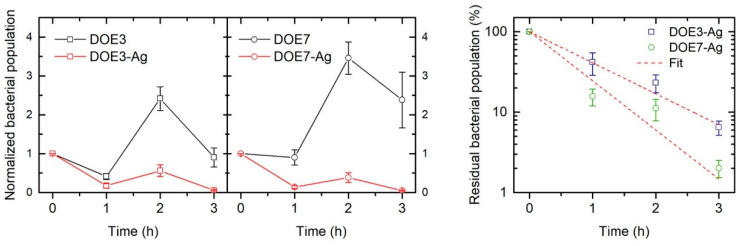
Left panel: Bacterial population on uncoated (black points) and coated (red points) samples normalized with respect to the *E. coli* control culture as a function of time. Right panel: Percentage residual bacterial population on the Ag-coated samples with respect to the corresponding uncoated samples for DOE3 (blue squares) and DOE7 (green circles) as a function of time, together with exponential decay fit (dotted red lines).

**Table 1 nanomaterials-12-03962-t001:** Inputs (*X_s_*)—selected processing parameters for the electrospinning of PSU solution 24% *w/w*.

	Parameter	Label	Unit	Low Level (−1)	High Level (+1)
*X* _1_	Flow Rate	FR	mL/h	1	2
*X* _2_	Voltage at Injector	Vi	kV	5	10
*X* _3_	Working Distance	d	cm	16	18

**Table 2 nanomaterials-12-03962-t002:** Outputs (*Y_s_*)—selected product properties of the electrospun PSU membrane.

	Parameter	Unit	Label
*Y* _1_	Ultimate Tensile Strength	MPa	UTS
*Y* _2_	Young modulus	MPa	E

**Table 3 nanomaterials-12-03962-t003:** DOE summary table, with the eight runs reported in standard order with details of corresponding values for inputs (*X_s_*) and outputs (*Y_s_*).

Standard Order	Sample ID	*X_s_*			*Y_s_*	
		FR (mL/h)	Vi (kV)	D (cm)	*Y*_1_: UTS (MPa)	*Y*_2_: E (MPa)
1	DOE1	1	5	16	4.5	291
2	DOE2	1	10	16	2.5	82
3	DOE3	2	5	16	18.5	805
4	DOE4	2	10	16	8.3	309
5	DOE5	1	5	18	3.0	157
6	DOE6	1	10	18	9.5	529
7	DOE7	2	5	18	12.0	459
8	DOE8	2	10	18	8.6	411

**Table 4 nanomaterials-12-03962-t004:** Coefficient of determination for fitted models outputs (*Y_s_*).

Best Models for PSU	R-sq	R-sq (adj)	R-sq (pred)
*Y*_1_: UTS (MPa)	99.78%	99.24%	96.54%
*Y*_2_: Young Modulus (MPa)	99.22%	97.26%	87.47%

**Table 5 nanomaterials-12-03962-t005:** ANOVA results.

Source		DF	Adj SS	Adj MS	F-Value	*p*-Value
Model		5	194.976	38.9952	184.59	0.005
	Linear	2	107.652	53.8262	254.80	0.004
	FR (mL)	1	97.301	97.3012	460.60	0.002
	Vi (kV)	1	10.351	10.3513	49.00	0.020
	2-Way interactions	3	87.324	29.1079	137.79	0.007
	FR (mL) * Vi (kV)	1	40.951	40.9513	193.85	0.005
	FR (mmL) * d (cm)	1	17.111	17.1113	81.00	0.012
	Vi (kV) * d (cm)	1	29.261	29.2613	138.51	0.007
Error		2	0.423	0.2113		
Total		7	195.399			

**Table 6 nanomaterials-12-03962-t006:** Bacterial population (CFU/mL) measured for control, uncoated samples (DOE3 and DOE7), and coated samples (DOE3-Ag and DOE7-Ag) at different times. Data are reported as mean value ± SD from two independent experiments, each in triplicate.

	Bacterial Population (CFU/mL)		
Treatment	1 h	2 h	3 h
Control *(E. coli)*	(8.9 ± 1.5) × 10^5^	(7.5 ± 0.9) × 10^5^	(7.3 ± 1.6) × 10^6^
DOE3	(3.5 ± 0.2) × 10^5^	(1.8 ± 0.9) × 10^6^	(6.6 ± 1.0) × 10^6^
DOE3-Ag	(1.5 ± 0.5) × 10^5^	(4.2 ± 1.0) × 10^5^	(4.2 ± 0.5) × 10^5^
DOE7	(1.3 ± 0.1) × 10^6^	(2.4 ± 0.1) × 10^6^	(1.7 ± 0.3) × 10^7^
DOE7-Ag	(2.1 ± 0.5) × 10^5^	(2.9 ± 0.8) × 10^5^	(3.5 ± 0.5) × 10^5^

## Data Availability

Not applicable.

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
