# Peer review of "Design and Manufacturing of Antibacterial Electrospun Polysulfone Membranes Functionalized by Ag Nanocoating via Magnetron Sputtering"

_nanomaterials, 2022, doi:10.3390/nano12223962_

Round 1

Reviewer 1 Report

1. The amount of Ag per 1 g of PSU fibers should be given, better from the independent analysis data. Are you sure that the total amount of Ag was the same in DOE3-Ag and DOE7-Ag samples?

2. “As mentioned in Section 184 2.2, membrane thickness was also evaluated via SEM on the sample cross section.”

- As you have already made a cross section, it would be reasonable to visualize by EDX map a spatial distribution of Ag inside the membrane, as the penetration depth of the sputtered Ag is rather limited.

3. Figure 7. Please explain briefly why the distribution of sulfur in the PSU-based material is not uniform. Why the density of sulfur dots in the element distribution map is much less than the corresponding density of carbon dots?

4. A magnification in Fig. 6b and Fig. 7(Ag) is comparable. A comparison of these micrographs shows that the Ag nanoparticles represent only a minor part of the particulated coating of PSU fibers. What is the composition of other nanoparticles? Why the non-metallic part of this coating is also particulated?

5. Figure 10. Please explain briefly the possible reasons of the hydrophobicity increase by the hydrophilic nanoparticles of Ag metal while the roughness of fiber surface of the Ag-coated and uncoated PSU fibers remains almost the same (see Fig. 6,7).

6. What was the reason to measure the hydrophobicity of these samples? Is it connected with their antimicrobial activity or with something else?

Author Response

We thank the reviewer for comments and suggestions.

Please find enclosed our reply and the revised manuscript.

Best regards

Reviewer 2 Report

The manuscript entitled “Design and manufacturing of antibacterial electrospun poly- sulfone membranes functionalized by Ag nanocoating via magnetron sputtering” describes the preparation of sulfone membranes functionalized by Ag nanocoating. This reviewer recommends the publication of this work after addressing the following comments.

Abstract, the outstanding point should be emphasized. Recently, antibacterial electrospun poly-sulfone membranes functionalized by Ag nanocoating have been already reported. Therefore, the outstanding point or different point of view need to be highlighted.

The author should provide all the purchased materials in the material and methods section.

Figure 8. FTIR spectra of (a) DOE3 and (b) DOE7 samples before and after Ag deposition. The author should perform the this FTIR studies again. There is no different in this study before and after Ag deposition.

Figure 9. X-ray diffraction patterns of (a) DOE3 and DOE3-Ag; (b) DOE7 and DOE7-Ag. What is the difference DOE3 and DOE3-Ag and DOE7 and DOE7-Ag in this XRD studies?

Conclusion should be revised to show the outstanding point of this work.

Author Response

(The authors gave the same response as above.)
